# Predicting Injuries in Football Based on Data Collected from GPS-Based Wearable Sensors

**DOI:** 10.3390/s23031227

**Published:** 2023-01-20

**Authors:** Tomasz Piłka, Bartłomiej Grzelak, Aleksandra Sadurska, Tomasz Górecki, Krzysztof Dyczkowski

**Affiliations:** 1Faculty of Mathematics and Computer Science, Adam Mickiewicz University, 61-614 Poznań, Poland; 2KKS Lech Poznań, 60-320 Poznań, Poland

**Keywords:** injury prediction, sport data analysis, rule-based system, expert system, fuzzy rule-based method, external training load

## Abstract

The growing intensity and frequency of matches in professional football leagues are related to the increasing physical player load. An incorrect training model results in over- or undertraining, which is related to a raised probability of an injury. This research focuses on predicting non-contact lower body injuries coming from over- or undertraining. The purpose of this analysis was to create decision-making models based on data collected during both training and match, which will enable the preparation of a tool to model the load and report the increased risk of injury for a given player in the upcoming microcycle. For this purpose, three decision-making methods were implemented. Rule-based and fuzzy rule-based methods were prepared based on expert understanding. As a machine learning baseline XGBoost algorithm was considered. Taking into account the dataset used containing parameters related to the external load of the player, it is possible to predict the risk of injury with a certain precision, depending on the method used. The most promising results were achieved by the machine learning method XGBoost algorithm (Precision 92.4%, Recall 96.5%, and F1-score 94.4%).

## 1. Introduction

Every year expectations of the performance of professional football players are growing. Increasing the number of high-intensity runs, accelerations, and decelerations and as the result growing overall player load, leads to a higher risk of occurrence of an injury [1]. Each exclusion of a player due to an injury may affect not only his individual physical condition and health but also the performance of the entire team [2]. In professional football, each won or lost match may decide whether a team moves to the next phase of the tournament or relegation from the league [3]. Injuries can have a significant impact on every level of a football club, hence minimizing the risk of their occurrence has become one of the most important tasks of their research departments. This research focuses on predicting non-contact lower body injuries resulting from overtraining or undertraining, as they can be predicted using decision-making models based on data including external training load, internal load, wellness data, etc. [4,5].

Prediction of injuries remains a difficult problem due to the individual biological differences of the body, different physical predispositions, or psychophysical condition of each player. Generalizations made by decision-making models can eliminate important factors influencing the actual condition of a given player’s health and in consequence the risk of injury. The aim of the project described in this paper is to use decision-making models to indicate the level of risk of injury with the highest possible accuracy. The research conducted, together with the individual motor profile of the football player created and updated, is used to determine his physical capabilities of the training load calculated at his individual level, which is used by physical preparation coaches. The model created, in view of the range of data, is general, it would be worthwhile in further research to expand it taking into account the individual predispositions of football players. The research being conducted is in response to a real problem occurring in football clubs. The club KKS Lech Poznań in the autumn round of the 2020/2021 season combined competition in both the domestic league (PKO BP Ekstraklasa), the European UEFA Cup competition, and the Polish Cup. In addition, some of the players were called up for the national teams of their countries held before the resumption of the games as well as during the autumn round. The intensity of the games has forced matches to be played both on weekends and in the middle of the week. As a result of such a busy calendar, limited recovery time between games, very short breaks between seasons, and rounds of individual games significantly increase the importance of training load management. External factors, such as long trips to other countries, often with a different time zone, and the participation in the training of those who play in the basic 11 or are substitutes, add to the burden. Those who have played fewer games or minutes in matches need a different training load to maintain proper form [6]. In the fall round preceding the testing period, the team suffered more than 30 different injuries over a five-month period, where the accumulation occurred in early autumn after the team played with a frequency of playing three games in two weeks.

This research aims to create a decision-making model to predict the probability of a non-contact injury coming from over or under-training among professional male football players. The preparation of such a model will enable a creation of a training load monitoring tool. Received reports from the tool can help the motor preparation team to adjust training loads given to an individual player to minimize the risk of occurrence of an injury in the next microcycle.

For this purpose, three decision-making models were created and compared. In the first phase, in cooperation with the motor preparation team, a rule-based method was prepared. Later on, a fuzzy rule-based system was prepared on the basis of expert knowledge-based rules. As a baseline for the machine learning method XGBoost algorithm was taken.

## 2. Related Work

Current studies in the publications deliver an insufficient understanding of what mostly affects injury risk, while an evaluation of the potential of statistical models in forecasting injuries is still missing. Based on research conducted so far, too high and too low training load leads to a higher risk of an injury occurrence [7]. An additional risk factor is changes in training load volume in microcycle’s series [4]. Both internal and external load parameters are correlated with the risk of an injury occurrence, however, some of them are more impactful, thus feature selection is one of the most important steps in this kind of experiment. In [8] authors presented a systematic review of methodology in practical injury prevention based on Acute: Chronic Workload Ratio. It compares different approaches to calculating the ACWR, e.g., using a rolling average, coupling methods, or EWMA (Exponentially Weighted Moving Average), which sets an increased weighting to the more recent workload values. Some of the compared studies show a relationship between increased ACWR and the risk of injury. However, the finally adopted methodology depends on the specificity of the research and its limitations. On the other hand, [9] presents a comparison of studies on injury risk assessment using elements of machine learning. This research concerns football players from different leagues, and seasons and uses a different set of parameters, e.g., external and internal load parameters. According to this research, machine learning does not seem to have high predictive power in every setting, nevertheless, it can support the identification of early symptoms of raised risk for an injury. Both reviews show different approaches to the issue of injury prediction and make it possible to refer to our approach, which is distinguished by the use of both machine learning methods, fuzzy systems, and a decision-making model based on expert knowledge. Our study is also characterized by limitations related to the dataset, therefore it uses only parameters related to external loading.

### 2.1. Acute: Chronic Workload Ratio

Acute: Chronic Workload Ratio is a parameter often used in modeling training load. In theory, there is a range of ACWR in which the risk of injury is increased, while research shows that it should not be used in isolation to analyze the causality between load and injury. In [10] authors investigate whether acute workload and chronic workload predict injury in professional rugby league players. Data were gathered from 53 rugby players during two league seasons. The acute workload is understood as a 1-week total distance and the chronic workload is a 4-week average acute workload. The acute: chronic workload ratio was computed by dividing acute workload by chronic workload. Considering acute and chronic workloads in isolation did not always predict injury possibility. Higher workloads can positively or negatively affect injury possibilities in rugby league players. Unlike players with a low chronic workload, players with a high chronic workload are more vulnerable to injury with moderate-low through moderate-high acute: chronic workload ratios. However, they are less vulnerable to injury in case of spikes in acute workload, so when the acute chronic workload is very high.

### 2.2. Building Multi-Dimensional Injury Forecaster

In the [5] a more complex and effective approach to injury prediction was presented. In this research, they present a multi-dimensional model for injury prediction in professional soccer using machine learning methods. Data were collected using GPS devices among players in a professional soccer organization during a season. The dataset consists of information about the 12 workload attributes extracted from the GPS data and the 6 personal attributes, 12 attributes computed as the Exponential Weighted Moving Average of the 12 workload features, 12 features consisting of the ACWR of the 12 workload attributes, 12 attributes consisting of the monotony of the 12 workload attributes and previous injury feature. They implemented the decision tree classifier and compared its results with the random forest classifier, the logit classifier, the four baselines, and the ACWR- and MSWR-based forecasters. The best results gave a decision tree classifier with recall 0.80±0.07 and precision 0.50±0.11 on the injury group, indicating that the decision tree classifier can forecast almost all the injuries and that it correctly labels a workout session as an injury in 50% of the cases.

## 3. Materials and Methods

### 3.1. Data Collection and Feature Selection

Data used in this research was collected during two rounds (spring round of season 2020/2021 and fall round of season 2021/2022) in the domestic football league (PKO PB Ekstraklasa), among 36 players with an average age of 24 (±5.26 SD). Measurements are made of players of all positions, excluding goalkeepers. The data were gathered using Catapult wearable global positioning trackers [11], both during exercise and game activities.

The data were collected using Catapult GPS technology sampling at 10 Hz (Vector S7 4 GHz, Catapult Innovations, Melbourne, Australia), which provided information on the players’ movement activities during a training session or match encounter. The GPS device, such as those placed in Figure 1, also included a tri-axial accelerometer, gyroscope, and magnetometer sampling at 100 Hz. According to the manufacturer’s assurances and certification, the device provides reliable and credible measurements during both open-air training sessions and matches in the stadium [11].

During all training sessions included in the study, each athlete used exactly the same data collection device. The devices were placed in dedicated vests sized to fit the athlete so as not to affect the exercises performed during training. The player, wearing such a vest during training, is shown in Figure 2.

The Catapult system allows the analysis of the players’ performance in individual parameters in real-time, using a data receiver, or after a training session, the data are ripped using the dedicated Catapult OpenField Console software. At this stage, coaches mark individual training sessions and clean the data to training time ranges. Once this is done, the data are sent to a central application maintained in the cloud and additionally to a dedicated database. This database also stores information from medical reports. The process is shown in Figure 3.

All injuries are reported by players to the medical team and then documented in a medical report. This document contains detailed information about each injury, e.g., damaged tissue, injury date, injury mechanism, etc. After selecting the injuries considered in this project, that is, non-contact lower body injuries, each event from the main dataset has been labeled (1—if the player got injured, 0—if the player did not get injured) using information from the medical report.

The dataset contains information about 1064 events. From all the parameters provided by the Catapult system, only 7 have been selected. Each event was described by parameters listed in Table 1. The choice of attributes resulted from the experience of physical preparation coaches in analyzing the relationship between the external training load of athletes and its impact on the occurrence of non-contact muscle injuries. Attributes were included here, the analysis of which makes it possible to determine the intensity of training, looking also at parameters that generate a high metabolic load—running at speeds above 19.8 km/h, acceleration, and deceleration above 2–3 m/s. Research on similar datasets, also augmented with external data, was conducted by [12]. The data for each microcycle were split into two subsets. The first aggregates activities from the entire workout week to game day, while the second aggregates match day results. The dataset was divided into train and test, containing 693/36 and 371/31 events/injuries respectively.

The dataset also contains similarly aggregated data as in the case of parameters related to microcycles, e.g., mc_TotalPlayerLoad, mc_FieldTime, etc, but for 1, 2, and 3 microcycles back. This provides insight into the training loads incurred in previous cycles that may impact the risk of injury. For each microcycle-related parameter, it is then e.g., mc_TotalPlayerLoad, mc_TotalPlayerLoad_-1, mc_TotalPlayerLoad_-2, mc_TotalPlayerLoad_-3, etc.

Together with the physical preparation coaching staff, the motor profile of the players was developed. An analysis was made of the players’ loads in terms of distance covered during the match, Sprint Distance, High Sprint Running (HSR), braking, and acceleration. Activities from matches in which a given player played for at least 75 min were used to determine reference values in the profile. For players who played fewer minutes in matches, the profile is the median of the results achieved by other players playing the same position. Based on the principles of creating a physical profile of football players, a set of additional variables was prepared, which in further processing was the basis for building a based-rule system.

Each of the variables defined is an expression derived from data collected from GPS receivers. For example, the variable REG_HSR_R1_A can be expressed as:IF [mc_HSR_-1] <> 0:    THEN REG_HSR_R1_A = ([mc_HSR]+ md_HSR]) / ([mc_HSR_-1] + [dc_HSR_-1])    ELSE REG_HSR_R1_A = 1.1

### 3.2. Expert Knowledge-Based Rules

In addition to the features listed in Table 2 and the features related to microcycles for previous microcycles, the dataset also contains rule values established on the basis of expert knowledge in cooperation with physical preparation coaches.

Any rule consists of two sections: the IF section named the antecedent (premise or condition) and the THEN section named the consequent (conclusion or action).

The fundamental syntax of a rule is:IF <antecedent>THEN <consequent>

In general, a rule can have numerous antecedents joined by the keywords AND (conjunction), OR (disjunction) or a mix of both [13].

In our model, six expert rules were prepared based on the expertise of the coaching staff and a review of the literature where they considered what the training load should be in the microcycle relative to match performance [4,14]. An example rule returning a value for ACWR is displayed below:## Calculation of returned values by ACWR ruleIF    (REG_ACWR  >= 0.80) AND (REG_ACWR <= 1.30)THEN RULE_ACWR = 0
IF    ((REG_ACWR  >= 0.70) AND (REG_ACWR  < 0.80))   OR ((REG_ACWR  > 1.30) AND (REG_ACWR  <= 1.40))THEN RULE_ACWR = 3
IF    ((REG_ACWR >= 0.50) AND (REG_ACWR < 0.70))   OR ((REG_ACWR > 1.40) AND (REG_ACWR <= 1.60))THEN RULE_ACWR = 7
IF    (REG_ACWR > 1.60) OR (REG_ACWR < 0.50)THEN RULE_ACWR = 10

In the rule-based approach, after each rule calculates a value, a final decision is made. It is assumed that all rules are equivalent, and the result is formed in the results aggregation of their values. Various approaches were tested, and it was decided to use average functions as aggregation. From the values returned by the rules, the average is calculated, and then two answers are generated: 0 - if the average was less than 6.5, 1 for a value at least equal to 6.5. In conducting the experiments, several approaches were tested to determine the final value before the rule system, including the maximum, the most frequent value returned by individual rules. The experiments conducted gave the best results for the described approach. A diagram of the data processing process is shown in Figure 4.

However, further work in the project showed that it was necessary to consider modifying some of the rules or giving up some of the additionally developed variables.

### 3.3. Fuzzy Rule-Based Model

The use of fuzzy set theory [15,16] and the fuzzy rule-based model [17,18] allows a more flexible interpretation of the input data. In this model, linguistic variables are defined that allow the imprecision of the input data to be taken into account and the sharp boundaries of the decision intervals from the classical expert rule-based decision model to be blurred. Such models are widely used when the input data may be subject to measurement error or the boundaries of the decision intervals cannot be strictly defined. Moreover, this approach allows the definition of the membership of given attributes to multiple classes at once with different degrees. The principles and examples of the fuzzy rule-based controller (decision system) can be found in [18,19,20].

The fuzzy rule-based model was implemented using the *Simful* Python library for fuzzy logic [21]. Two approaches have been used to build the decision system. In the first one, the rules formulated by specialists were used (the model presented in the previous section was used), in the second the variables and rules were inducted on the basis of available data using data analysis and clustering methods (the *Pyfume* package [22] was used for this purpose). Finally, the combination of the two approaches resulted in the decision model presented next.

For each attribute of input and output data, a linguistic variable with terms in the form of fuzzy sets is constructed. Each variable has its own range and a corresponding list of terms (fuzzy sets), which are then used in decision rules. Figure 5 shows a selection of input variables and an output variable. All definitions of the input linguistic variables prepared in the model can be found in the supplied code.

The model is based on 34 rules. Example rules are presented below. The full set of rules is provided in the supplied code.IF (HSR_R1_A IS small) AND (HSR_R1_B IS small) AND (HSR_R1_C IS small)   THEN (load IS very_small)IF (REG_ACWR IS small) OR (REG_ACWR IS big) THEN (load IS small)IF (HSR_R2 IS big) THEN (load IS big)IF (SPRINT_R1_A IS small) AND (SPRINT_R1_B IS small) AND   (SPRINT_R1_C IS small) THEN (load IS very_small)IF (REG_ACC IS small) THEN (load IS small)

The system returns the training load decision as a real number in the range [0,10], which is interpreted as the chance of an injury occurring. A level of 0.6 was used as a cut-off point, above which the system interprets the value as an injury occurs.

### 3.4. Machine Learning Model

As a machine learning model, the gradient boosting method (XGBoost) was used [23]. It is a machine learning method, which delivers a model in the form of an ensemble of weak models, commonly decision trees. It creates the model like other boosting methods, but generalizes them, allowing the optimization of an arbitrarily differentiable loss function.

The XGBoost algorithm was implemented using the R language. The *caret* package [24] was used to perform cross-validation and hyperparameter tuning using the grid search technique. To inspect the quality of the XGBoost model 5 × 10 cross-validation method was applied. The selected model consists of 500 trees. The exact values of all tuned hyperparameters can be found in Table 3.

Since the training dataset contains 657 non-injury observations and just 36 injury observations, class distribution is highly unbalanced. To adjust this imbalance the minority group in the training dataset was oversampled by using Synthetic Minority Oversampling Technique (SMOTE) [25].

As the design of the microcycle data means that there were gaps in the data, these were filled by simple imputation using the median. We also tested a much more sophisticated Multivariate Imputation by Chained Equations (MICE) [26] algorithm, but with no improvement in quality.

## 4. Results

To compare models we used popular methods of evaluating classification tasks:**Accuracy**: Percentage of correctly classified cases.**Precision**: Ratio of correctly classified items in the ”injury” class to all that the classifier has marked as “injury”.**Recall**: The ratio of correctly recognized elements from the ”injury” class to all the elements it should recognize, i.e., the entire “injury” class.**F1**: Harmonic average of precision and recall.

The results for each of the models are different (for all models we had the same test dataset). There is still room for improvement in the rule-based and fuzzy rule-based models. The best results achieved on the test set by each model were shown in Table 4.

The fuzzy model showed the highest number of false positive results, so 65 non-injury events were classified as injuries. False positive results are particularly undesirable in this study because they can lead to unreasonable prevention of the player from participating in the match or training session and should be minimized in further work. The fuzzy rule-based method achieved Accuracy = 0.84, Precision = 0.16, Recall = 0.23 and F1-Score = 0.19.

The rules created for the decision model based on expert knowledge allowed us to obtain satisfactory prediction results, achieving Accuracy = 0.76, Precision = 0.53, Recall = 0.58, and F1-Score = 0.53. The rule-based model classified 21 injuries correctly, however, 1 injury event was undetected. The model predicted 36 events as false positive results.

The model using machine learning methods gave the best results for injury prediction. We prepared three models which differ train dataset used to construct:All features and additional rules as features.All features without our rules as features.Only our rules as features.

As presented in Table 5, the best XGBoost algorithm achieved Accuracy = 90.0, Precision = 92.0, Recall 97.6 and F1-Score = 94.7.

For further analysis, a feature importance plot (importances are scaled to have a maximum value of 100%) for the best XGBoost model presented in Figure 6 was generated. Feature importance is measured as the gain contribution of each feature to the model. A higher percentage means a more important predictive feature.

Analyzing the relevance of individual attributes, it should be noted that the model indicated a high significance of four attributes: the amount of time the athlete spent training, the two weeks prior to the microcycle analyzed, the number of decelerations and accelerations, and the number of meters run at a speed of 19.8–25.2 km/h (HSR) during the current training microcycle. These features were confirmed by physical training coaches. Particular attention was paid to the importance of the number of accelerations and braking. In multiplayer games with such a high number of contacts between players, and with frequently changing dynamics and directions of movements, it is these two parameters that should be monitored. This has been implemented in training monitoring, what will be described in the last section of the article. The greatest relevance returned by the model relates to the player’s total training time, but in a microcycle that took place two microcycle units before the training microcycle under analysis. This is consistent with the ACWR approach. When analyzing the training load, it is important to consider not only the current and previous microcycle, but also to reach back to the athlete’s previous activities. This is in line with the principles of developing the team’s training, where training is supposed to have an effect not only in the short term but also in terms of the next few weeks. It is worth noting that the model indicated in the features a high validity of the activities obtained two microcycles before the current microcycle. The elements indicated in the relevance of the features largely coincide with the analyses conducted on player injuries and training load in the microcycles preceding the onset of injury. However, in the case of a different approach to the aggregation of microcycle data, or the occurrence of interspersed microcycles of different long (6–7 days) and short (2–3 days) lengths during the season, other attributes may be proposed for consideration. The inhibition parameter was most often a missing training element in players who did not play at least 45 min in a match and difficult to implement in compensatory training, where one tries to achieve load values in individual attributes close to the results of a player’s match profile. Therefore, measures have been taken to stimulate muscles through eccentric training from benchmark braking moments on external strength equipment. This will be used to evaluate the model in further research.

In addition, Figure 7 was presented to check the quality of our rules.

## 5. Discussion

All the developed models provide a certain perspective on the risk of injury and can be used to provide early alarms about incorrectly selected training loads. During the analysis of the results obtained, in particular, in the case of the rule-based decision model, a tendency was noticed to detect an injury in the microcycle preceding the actual injury. Therefore, the development of a fuzzy measure of the quality of the model remains to be considered, which will allow not so much to indicate a specific microcycle, but a selected wider area at risk of injury.

What is worth pointing out, the model using machine learning methods achieved very good effectiveness in injury detection. This approach may therefore be considered appropriate in the case of such a complex and multidimensional problem as injury prediction. The XGBoost algorithm’s false positive rate is small, indicating that it reduces the “false alarms”. In professional football, false positives are highly undesirable as an unreasonable absence of players can negatively affect a team’s performance. The machine learning model also generates a moderate false-negative rate, which means that it is rare for an injured player to be classified as out of the injury-risk situation.

### Future Research Directions

Despite the satisfactory results obtained in this study, there are still many directions for future development. First of all, the dataset could be enriched with factors related to internal loads, such as heart rate during activity, Rate of Perceived Exertion [27] etc. A wider range of data on the fitness and quality of training of a given athlete may improve the quality of inference without interfering with the construction of the model. However, to collect more data, additional arrangements should be made with the club as this may be related to the need to provide additional measuring equipment. Moreover, the introduction of data on previous injuries to the training set may have a significant impact on the results obtained.

Another proposed approach to improving the quality of injury prediction is to aggregate the results of all three developed models and to generalize the final prediction score. The combination of systems based on expert knowledge and using elements of machine learning may improve the results of prediction.

The final task will be to prepare a tool to assess the planned training load in the next microcycle and its impact on the risk of injury. It will be a significant improvement from the club’s point of view, which will allow for a better adjustment of the training model, and thus improve sports results and reduce the number of injured players over the course of the season. As injuries have a high economic cost for the club, due to the costly process of convalescence and rehabilitation of players, preventing injuries can reduce these costs, which means enhancing the team’s performance and mental state of the players, as well as reducing seasonal medical expenses [28].

The problem of injury prediction is extremely important in the context of appropriate training intensities and volumes, as well as the preparation of football players to perform in matches. Studies have shown that it is necessary to specifically control the appropriate level for each player of training the number of meters run at appropriate speeds, as well as acceleration and deceleration. This issue is particularly important when a team participates in several different games during the season. Then matches are held both on weekends and during the week and the sheer amount of time spent preparing for the next match is reduced. It is then important to train gear in terms of HSR, braking, and acceleration three days before the next match meeting. Load control already takes place during matches, players who do not play in the basic lineup in a match should have compensatory training to equalize their loads and ensure readiness for the rest of the matches. Using straight-line running in compensation training, we are able to approach the match profile in the speeds generated by the players, while the most difficult element in training is the braking element. It is recommended as a complement to compensatory training, an eccentric force training session using machines that provide concentric, eccentric, and isometric resistance resulting in explosive maximal force development and delayed eccentric loading. This approach was used at the club, which will be taken into account in future iterations of the work on the injury prediction issue. The research is based on real data, by which it is impossible to fully simulate it in order to confirm the validity or negation of the experiments conducted. The verification criterion adopted in the conducted study, in cooperation with the club, is the analysis of training loads taking into account the individual capabilities of the players and the proposed models of analysis in subsequent seasons, along with the analysis of the number and type of non-contact muscle injuries occurring in the players. 

## Figures and Tables

**Figure 1 sensors-23-01227-f001:**
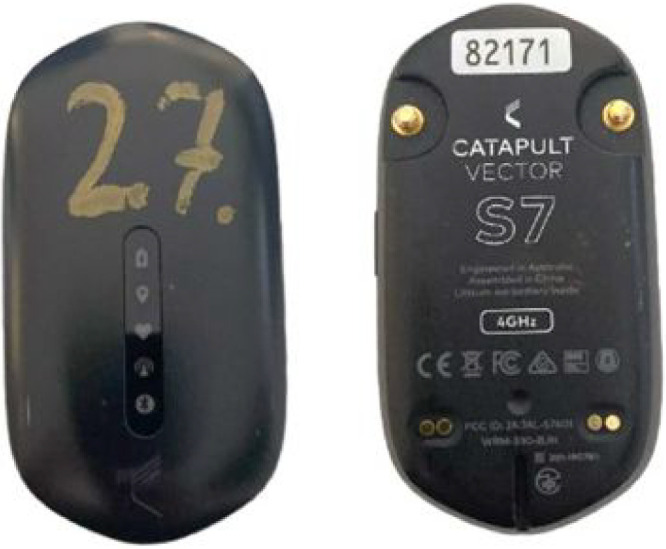
Catapult wearable device for data collection used at the club.

**Figure 2 sensors-23-01227-f002:**
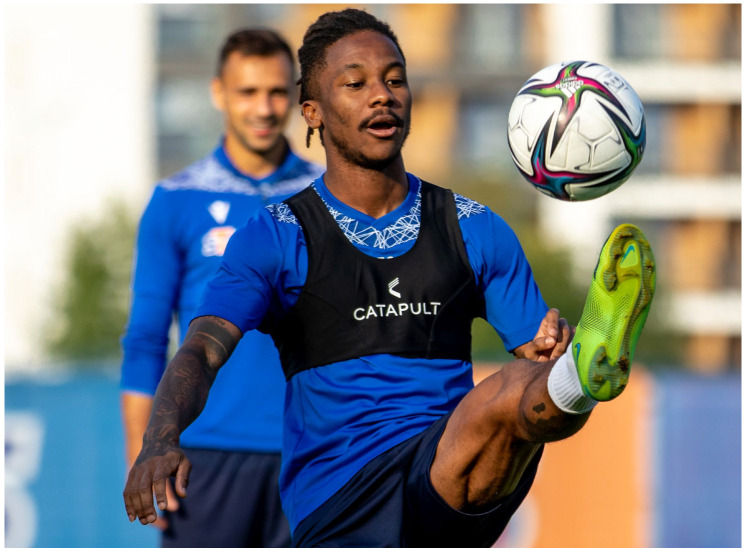
A player wearing a receiver vest during training, photo by Marcin Rajczak, KKS Lech Poznań.

**Figure 3 sensors-23-01227-f003:**
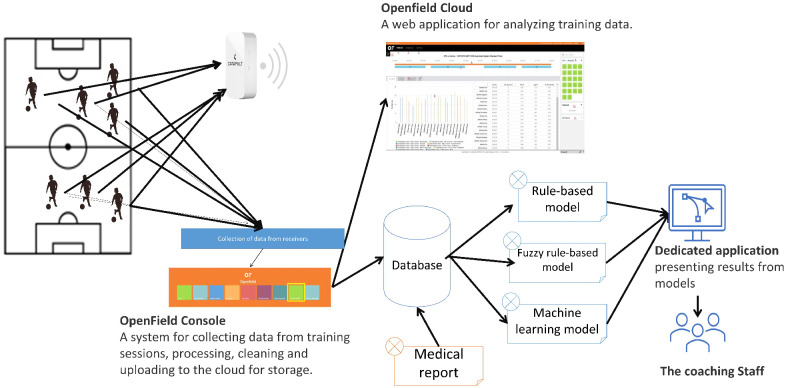
The process of collecting and processing training activity data from Catapult GPS wearable retrievers.

**Figure 4 sensors-23-01227-f004:**
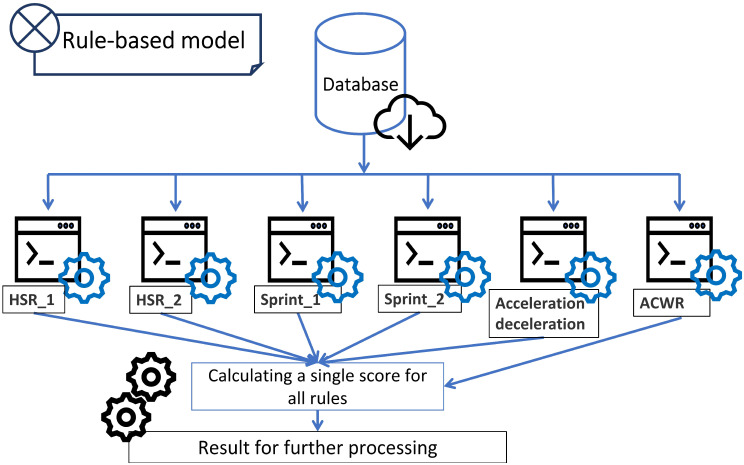
Processing in a rule-based system.

**Figure 5 sensors-23-01227-f005:**
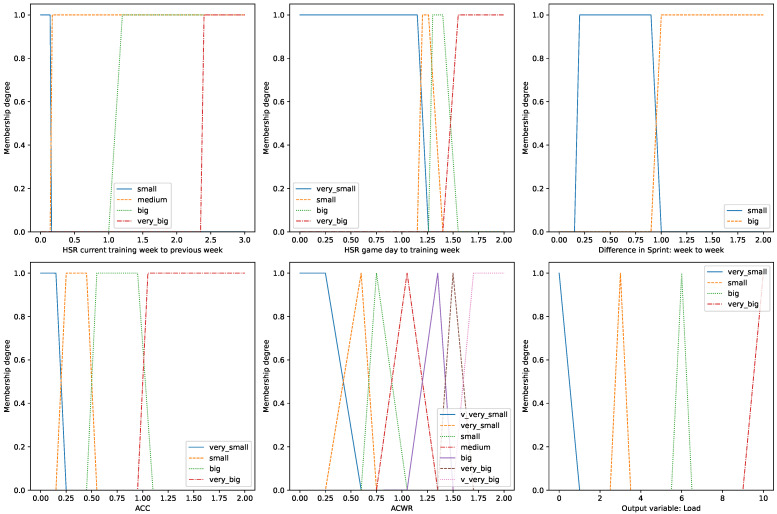
Selected linguistics variables in fuzzy rule-based model.

**Figure 6 sensors-23-01227-f006:**
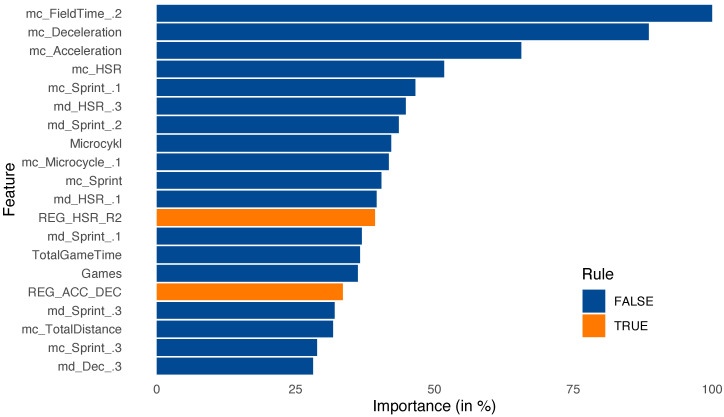
Feature importance (20 most important features) generated from machine learning model (XGBoost (1)).

**Figure 7 sensors-23-01227-f007:**
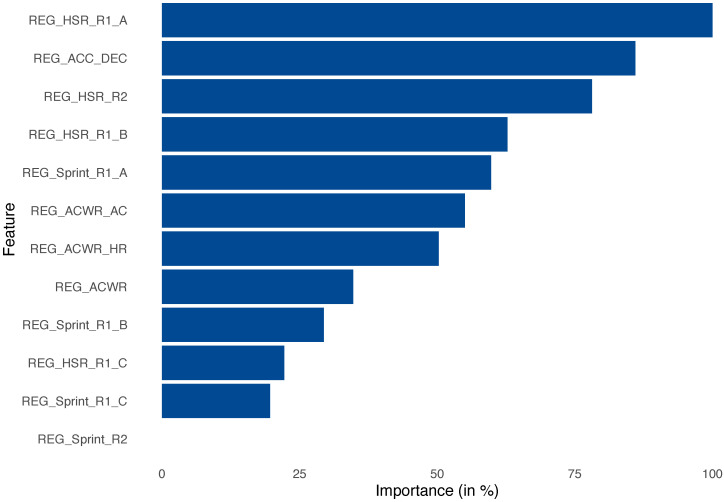
Feature importance generated from machine learning model (XGBoost (2)).

**Table 1 sensors-23-01227-t001:** Basic parameters used in the dataset.

Parameter	Description
PlayerId	Player Id
PlayerPosition	Position of the player
mc_Microcycle	Number of microcycle in season
Injury	1—Yes, 0—No, if the injury occurred in a microcycle
TotalTrainingTime	Sum of minutes of training which the player has participated in since the beginning of the round (preseason included)
TotalGameTime	Minutes of play in previous games in season
Games	Number of games played before each workout session
PlayInMatch	1—Yes, 0—No, if the player plays in a match during the analyzed microcycle
Sum of activity in training microcycle, excluding match day	
mc_TotalDistance	Total distance covered in training
mc_HSR	Distance in meters covered in HSR (19.8–25.2 km/h)
mc_Sprint	Distance in meters covered in Sprint (>25.2 km/h)
mc_TotalPlayerLoad	Player load
mc_FieldTime	Training time in microcycle
mc_Acceleration	Accelerations above 2–3 m/s${}^{2}$
mc_Deceleration	Decelerations above 2–3 m/s${}^{2}$
Activity in match day	
md_TotalDistance	Total distance
md_HSR	Distance covered in HSR (19.8–25.2 km/h)
md_Sprint	Distance covered in Sprint (>25.2 km/h)
md_TotalPlayerLoad	Player Load
md_FieldTime	Player game time
md_Acceleration	Accelerations above 2–3 m/s${}^{2}$
md_Deceleration	Decelerations above 2–3 m/s${}^{2}$

**Table 2 sensors-23-01227-t002:** Additional variables prepared for the system to determine the relationship between data from different macrocycles.

Rule	Description
HSR variables, for each ratio	
REG_HSR_R1_A	The sum of HSR values achieved in the current microcycle, to the values achieved in the microcycle preceding the microcycle under analysis.
REG_HSR_R1_C	The sum of the HSR values achieved in the microcycle three weeks before the current one, to the values achieved in the microcycle preceding the microcycle under analysis.
REG_HSR_R2	The HSR values achieved in a match, to the sum of the values achieved in a microcycle, without a match, in the microcycle analyzed.
Sprint variables, for each ratio	
REG_Sprint_R1_A	The sum of Sprint values achieved in the current microcycle, to the values achieved in the microcycle preceding the microcycle under analysis.
REG_Sprint_R1_C	The sum of the Sprint values achieved in the microcycle three weeks before the current one, to the values achieved in the microcycle preceding the microcycle under analysis.
REG_Sprint_R2	The Sprint values achieved in a match, to the sum of the values achieved in a microcycle, without a match, in the microcycle analyzed.
Acceleration and deceleration variables	
REG_ACC_DEC	The ratio of the sum of acceleration and deceleration achieved in a match, to the sum of values achieved in a microcycle, without a match, in the analyzed microcycle.
Acute: Chronic Workload Ratio variables	
REG_ACWR_AC	The sum of the PlayerLoad parameter values from the entire microcycle (match and training).
REG_ACWR_HR	The average value of the PlayerLoad parameter obtained in the current microcycle and the three preceding it from the entire microcycle (match and training).
REG_ACWR	Ratio of values of parameters REG_ACWR_AC and REG_ACWR_HR.

**Table 3 sensors-23-01227-t003:** Final hyperparameters of XGBoost.

Hyperparameter	Value
nrounds	500
max_depth	7
eta	0.05
gamma	0.01
colsample_bytree	0.75
min_child_weight	0.00
subsample	0.50

**Table 4 sensors-23-01227-t004:** The best classification results on test set for each model.

Model Type	Accuracy	Precision	Recall	F1
Fuzzy rule-based	83.6%	15.9%	22.6%	18.7%
Rule-based	76.0%	53.0%	58.0%	53.0%
Machine learning	90.0%	92.0%	97.6%	94.7%

**Table 5 sensors-23-01227-t005:** Machine learning method (XGBoost) classification results on the test set.

Approach	Accuracy	Precision	Recall	F1
(1)	90.0%	92.0%	97.6%	94.7%
(2)	86.8%	92.2%	93.5%	92.8%
(3)	78.7%	92.2%	83.8%	87.8%

## Data Availability

The data presented in this study are available on request from the corresponding author. The data are not publicly available due to privacy.

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
