# Peer review of "Predicting Injuries in Football Based on Data Collected from GPS-Based Wearable Sensors"

_sensors, 2023, doi:10.3390/s23031227_

Round 1
Reviewer 1 Report
The paper deals with decision-making models used to predict risk of non-contact injuries due to both over and undertraining in football. The data for the analysis was acquired by the Authors with wearable GPS sensing devices. There have been developed and assessed several models to model the load and determine the risk of injury (e.g., based on fuzzy sets, machine learning). Based on the obtained results, the capabilities for decision making models were discussed. The work stands for a comprehensive view on various specific factors that may influence occurrence of injuries. The proposed approach to „aggregate the results of all three developed models and to generalize the final prediction score” (announced in Section 5.1) seems to be a good choice for the future study. In the reviewer’s opinion the paper should be published after the raised comments are addressed by the Authors. This comments are provided below.
1) What is an objective (possibly general) criterion that may be used to assess the repeatability and reliability of the proposed and tested models considering their specificity (sport activity, player performance)?
2) What were the criteria used to the selection on the measurement data used for the set for models training?
Row 146: “The dataset contains information about 1064 events. From all the parameters provided by the Catapult system, only 7 have been selected.” Was something specific about that choice? Were the Authors looking for some specific type of injuries? Some comment on this choice should be added even if it was made arbitrarily.
3) Row 183: „In our model, six expert rules were prepared based on the expertise of the coaching staff and a review of the literature.” Is there any particular publication providing this data?
4) Row 203: “the average is calculated” - Is there any other approach to calculate the resultant value (weighted average, etc.)?
5) As reported in the manuscript, both undertraining and overtraining may disadvantageously influence the player’s overall performance. Hence, is there any Authors’ expectation to get some global minimum declaring the most desired training scenario for a player considering his or her differences of the body, different physical predispositions, or psychophysical condition?
6) How reliable (I mean repeatable) is the data presented in Figure 6? Is there any risk that the choice on the most contributing feature importance may be significantly influenced by the specificity of the measured data used for the analysis? In the other words, should we expect that other set of data will result in different view of plot in Figure 6? Can we reliably formulate general conclusions based on Figure 6? The same comment applies to Figure 7. A comment on that should be added to the manuscript. What should be the most influential factors, generally and irrespectively from the specificity of the analyzed data?
Minor comments:
Row 298: please use more formal form for “as we will tell you in the last section of the article.”
Row 307: please paraphrase “in the microcycle minus two to the current microcycle”
If no financial support was present, the section “Acknowledgments” should be removed or filled in considering the publisher requirements.
Row 306: What do “the liku features” mean?
Reviewer 2 Report
The motivation of the work is quite reasonable and interesting. I found some unclear points that need to be addressed before final judgment toward publication.
1) Were the test data same across the three methods, i.e., rule-based, fuzzy rule-based, and machine learning based. If not, the same test data should be used for fair comparison.
2) How robust against individual difference? I wonder if the test was carried out by k-fold cross-validation (in machine learning-based method), which contains the data from persons whose data are also in the training data. In this case, the performance metrics tend to be high because the trained model knows the information about the person in the test data. If not yet tried, I strongly suggest to test by "leave-one-person-out" cross-validation scheme to understand a practical performance of the classification method.
The followings are suggestions for improvement.
a) The definition of Accuracy, Precision, Recall, and F1-score could be presented.
b) The value "0.54" in line 269 should be 0.55 according to Table 4.
c) The term "data" should be handled as "plural". So, "was" in line 122 should be "were".
d) Figures 3 and 4 should be clearly presented, which are currently blurred.
